# Modulation of Endothelial Glycocalyx and Microcirculation in Healthy Young Men during High-Intensity Sprint Interval Cycling-Exercise by Supplementation with Pomegranate Extract. A Randomized Controlled Trial

**DOI:** 10.3390/ijerph17124405

**Published:** 2020-06-19

**Authors:** Zivile Pranskuniene, Egle Belousoviene, Neringa Baranauskiene, Nerijus Eimantas, Egle Vaitkaitiene, Jurga Bernatoniene, Marius Brazaitis, Andrius Pranskunas

**Affiliations:** 1Department of Drug Technology and Social Pharmacy, Lithuanian University of Health Sciences, Sukileliu pr.13, LT-50162 Kaunas, Lithuania; zivile.pranskuniene@lsmuni.lt (Z.P.); jurga.bernatoniene@lsmuni.lt (J.B.); 2Institute of Pharmaceutical Technologies, Lithuanian University of Health Sciences, Sukileliu pr.13, LT-50162 Kaunas, Lithuania; 3Department of Intensive Care Medicine, Lithuanian University of Health Sciences, Eiveniu g. 2, LT-50161 Kaunas, Lithuania; egle.belousoviene@lsmuni.lt; 4Institute of Sport Science and Innovation, Lithuanian Sports University, Sporto g. 6, LT-44221 Kaunas, Lithuania; neringa.baranauskiene@lsu.lt (N.B.); nerijus.eimantas@lsu.lt (N.E.); marius.brazaitis@lsu.lt (M.B.); 5Department of Disaster Medicine and Health Research Institute, Lithuanian University of Health Sciences, Eiveniu g. 4, LT-50161 Kaunas, Lithuania; egle.vaitkaitiene@lsmuni.lt

**Keywords:** microcirculation, glycocalyx, polyphenols, sprint exercise

## Abstract

The natural components of the pomegranate fruit may provide additional benefits for endothelial function and microcirculation. It was hypothesized that supplementation with pomegranate extract might improve glycocalyx properties and microcirculation during acute high-intensity sprint interval cycling exercise. Eighteen healthy and recreationally active male volunteers 22–28 years of age were recruited randomly to the experimental and control groups. The experimental group was supplemented with pomegranate extract 20 mL (720 mg phenolic compounds) for two weeks. At the beginning and end of the study, the participants completed a high-intensity sprint interval cycling-exercise protocol. The microcirculation flow and density parameters, glycocalyx markers, systemic hemodynamics, lactate, and glucose concentration were evaluated before and after the initial and repeated (after 2 weeks supplementation) exercise bouts. There were no significant differences in the microcirculation or glycocalyx over the course of the study (*p* < 0.05). The lactate concentration was significantly higher in both groups after the initial and repeated exercise bouts, and were significantly higher in the experimental group compared to the control group after the repeated bout: 13.2 (11.9–14.8) vs. 10.3 (9.3–12.7) mmol/L, *p* = 0.017. Two weeks of supplementation with pomegranate extract does not influence changes in the microcirculation and glycocalyx during acute high-intensity sprint interval cycling-exercise. Although an unexplained rise in blood lactate concentration was observed.

## 1. Introduction

The popularity of natural compounds to increase tissue perfusion and ergogenic effects during acute exercise is growing [1]. The natural components of pomegranate, including nitrate and polyphenols, may be of additional benefit for endothelial function [2]. Polyphenols modulate oxidative stress and inflammation, improve endothelial function, have vasodilatory effects, and promote nitric oxide (NO) production, all of which could be beneficial for increasing microvascular blood flow [2].

Previous studies have shown that supplementing with substances containing nitrates enhances exercise tolerance, mitochondrial efficiency, blood flow, and oxygenation via enhanced vasodilation and reduced vascular resistance [3]. Consuming pomegranate has been found not only to lower blood pressure but also to improve endothelial function in clinical populations such as hypertensive patients and have an ergogenic effect during high-intensity eccentric exercise in resistance-trained subjects [4,5]. Compared to the polyphenols contained in other fruits, those specific to pomegranate have demonstrated a significant positive effect on endothelial-dependent vasodilation [6,7,8]. The endothelium is the crucial component of the microvascular system, which ensures the local equilibrium between pro- and anti-inflammatory mediators, hemostatic balance, contributes to vascular permeability, and cell proliferation [9]. The central role in controlling vascular tone is played by NO, synthesized in the endothelium by the endothelial nitric oxide synthase isoform, which can be affected by shear stress, calcium, and O_2_ availability [10,11]. Constant shear stress maintains stable function of the vascular endothelium, preventing cell apoptosis and proliferation, clotting, leukocyte adhesion, and atherogenesis, while perturbed shear stress provokes shifts in excretion of vasodilator and vasoconstrictor substances [9]. An imbalance between reactive oxygen species (ROS) production and NO availability promotes major changes in endothelial homeostasis and leads to impaired microcirculatory perfusion [9].

It is widely known that regular training is beneficial for health and might help to prevent or delay cardiovascular, metabolic, and other chronic diseases [12]. However, acute and strenuous bouts of exercise become exhaustive and can induce ROS overproduction [12,13]. Endothelium-derived ROS contribute to oxidative stress upon exercised muscle and capillary damage [14] via inadequate oxygen supply and increased shear stress [15,16]. A vital element of microvasculature protecting capillaries from the physical impact of blood flow remains a sugar protein glycocalyx. It is a gel-like layer, that coats all healthy vascular endothelium on the luminal side, providing a micro-environment for many important vascular processes [17]. For example, glycocalyx regulates vascular permeability, controls blood cell–vessel wall interactions, and affects the rheology [17,18,19].

Unlike other sorts of high-intensity exercise, sprint interval exercise causes fatigue due to metabolic stress to muscle fibers with no injury. [20] This kind of training session usually includes a sequence of brief episodes of strenuous exercise separated by periods of rest or low-intensity activity [20]. A significant advantage regarding this is that beneficial adaptations can be achieved with much shorter exercise duration compared to endurance training [21].

Nevertheless, it is recognized that blood flow during intense exercise is redistributed towards the working muscle, cardiopulmonary system, and skin, diverting it from the splanchnic viscera [22,23]. It can provoke gastrointestinal ischemia-related symptoms such as nausea, vomiting, abdominal pain, and diarrhea, occasionally with blood stains [24]. There is data that maximal sprint effort, longer than 30 s, is linked to nausea, vomiting and dizziness [25]. As a part of the gastrointestinal tract, the sublingual area may be representative of changes in visceral microcirculation. Sublingual microcirculation is easily and noninvasively accessible and has been investigated more than other regions [26,27]. Furthermore, it is clinically relevant for identifying alterations of microcirculation and represents central microcirculation better than cutaneous perfusion [23,26,27]. Altered microcirculation in the sublingual region is related to worse outcomes in critical care patients, encountering a significant amount of oxidative stress [28].

The objective of the study was to investigate whether two weeks of supplementation with pomegranate extract may improve the glycocalyx characteristics and sublingual microcirculation during acute high-intensity sprint interval cycling exercises in young healthy men compared to those with no supplementation.

## 2. Materials and Methods

### 2.1. Experimental Design and Supplementation

A randomized controlled study was performed. At the beginning of the first experimental day, blood samples were drawn after 10 min of rest to measure the syndecan-1, glucose, and lactate concentration. Systemic hemodynamic parameters such as the mean arterial blood pressure (MAP), heart rate, and cardiac index (CI) were recorded using impedance cardiography (BioZ; CardioDynamics, San Diego, CA, USA) in the supine position. The microvascular flow and density parameters of the sublingual area were evaluated using incident dark field (IDF) video microscopy (Braedius Medical, Huizen, The Netherlands), and a glycocalyx image was obtained using a sidestream dark field (SDF) device (Microscan^®^; Microvision Medical, Amsterdam, The Netherlands).

The subjects first warmed up on the cycle ergometer for 10 min at a power output (W) approximately equal to the subject’s body mass (kg) and at a rate of 70 rpm.

Each sprint interval exercise (SIE) bout comprised 12 repeats of 5 s [29] on a cycle ergometer (Monark, Ergomedic 894 EA, Vansbro, Sweden) interfaced with a laptop computer. Straps were used to secure the feet to the pedals. The pedal right arm crank starting position was 45° forward to the vertical axis. Upon the start command, the subject started pedaling and proceeded until instructed to stop. After the 100 rpm pedaling frequency was reached, the computer automatically added the weight (7.5% of body mass). During the entire SIE, strong verbal encouragement was provided.

All measurements were repeated in the same manner under the same conditions immediately after the SIE. Additionally, power output and heart rate were registered during SIE.

The following morning, the participants in the pomegranate group began taking a pomegranate extract supplement daily for two weeks. Pomegranate supplement was obtained through a low heat reduction of pomegranate pulp, concentrated under vacuum, freeze-dried to preserve its properties, and then fermented. The extract contains the full spectrum of polyphenols out of the red fruit seeds and the white membranes of the seeds, which are typically not consumed. Liquid thick pomegranate extract was provided in a graded cup.

Each 20 mL dose (720 mg phenolic compounds measured as a gallic acid equivalent according to Folin-Ciocalteu) was administered personally at the university pharmacy daily at the same time (9–10 a.m.). The members of the control group were not given any substances.

After the two weeks of supplementation, all measurements were replicated before and after the SIE (repeated bout) in the same manner and under the same conditions described above (Figure 1).

### 2.2. Participants

Eighteen male volunteers from 22 to 28 years of age were recruited to participate in this study. Eligibility criteria for participants were as follows: healthy male, recreationally active, no history of smoking, or supplementation.

The participants were instructed to avoid exercise and the consumption of food containing large amounts of antioxidants or polyphenols during the study period. Each subject read and signed an informed consent form that was prepared according to the principles outlined in the Declaration of Helsinki, and the participants were randomly assigned to the pomegranate or control group (9 in each group) using online randomization tool (https://www.randomizer.org/). Table 1 outlines their baseline characteristics.

The protocol and consent form were fully approved by the Ethics Committee of Lithuanian University of Health Sciences and by the Kaunas Regional Biomedical Research Ethics Committee (number BEC-MF-02; BE-2-5).

### 2.3. Familiarization

Each subject was tested at the same time of day to reduce the effect of diurnal biological fluctuations [11]. On the familiarization day, all participants executed a maximum graded exercise to assess their maximum oxygen uptake (VO2max) on a cycle ergometer (Ergoselect 100; Ergoline GmbH, Bitz, Germany) using a mobile spirometry system (Oxycon Mobile; Jaeger/VIASYS Healthcare, Hoechberg, Germany). The ramp exercise test was employed to measure VO_2_max. This test consists of a 4-min warm up followed by a constant increase in the speed of 0.1 km/h every 6 s until fatigue. Three criteria were adopted to confirm that VO_2_max was achieved (all to be met): a respiratory exchange ratio greater than 1.1, a highest heart rate (HR) equal to 220 − age ± 10 beats per min, and a plateau in oxygen uptake with a rising workload [30]. Pulmonary gas exchange was analyzed using a portable analyzer (Oxycon Mobile; Jaeger, Hoechber, Germany). The equipment was calibrated following the manufacturer’s recommendations before each test. Taking into account the American College of Sports Medicine recommendation for individuals performing high-intensity exercise, only participants with a VO_2_max > 39.0 mL/kg/min were involved in the study [31]. The subjects were familiarized with all testing procedures, equipment, and the study protocol at least a week before the first day of investigation. Prior to each session, they refrained from coffee and alcohol for at least 24 h, avoided strenuous exercise for 72 h, and drank enough water to stay adequately hydrated.

### 2.4. Phenolic Compound Evaluation

An extract of partly fermented pomegranate juice and pulp (*Punica granatum* L.) was used. Detection of phenolic compounds was performed with the chemical reagents gallic acid (Sigma Aldrich, Germany), Folin–Ciocalteu reagent, and sodium carbonate (both from Fluka Chemie, Germany). Average nutritional values for 100 mL of pomegranate supplement were as follows: fat < 0.5 g, carbohydrate 47 g, protein 0.8 g, salt 0.04 g, potassium 1500 mg. 

The total amount of phenolic compounds was determined using a modified Folin–Ciocalteu colorimetry technique. The solution (0.7 mL) under study was placed into a 10-mL graded flask. The Folin–Ciocalteu reagent (400 μL) was subsequently added, and after 3 min, a sodium carbonate (Na_2_CO_3_) solution (75 g/L) was added. After 2 h, the suspension was centrifuged (5000 rpm for 5 min) and the supernatant was measured with a spectrophotometer at the 760 nm wavelength. The calibration curve was calculated with respect to gallic acid.

### 2.5. Evaluating Microcirculation

Video images of the sublingual microcirculation were captured using a handheld Cytocam IDF videomicroscope (Braedius Medical, Huizen, The Netherlands). This device is optimized to visualize microcirculation on organ surfaces. The principal of IDF imaging is based on the fact that the hemoglobin absorbs emitted green light (wavelength: 530 nm) and the red blood cells are therefore seen as black or gray bodies. The vessel walls cannot be visualized and thus are only detectable by the presence of red blood cells [32]. A recently published validation study demonstrated that Cytocam IDF imaging provides better image quality than SDF imaging [33]. By now, sublingual videomicroscopy has emerged as a gold standard for microcirculatory evaluation [34].

After gently removing saliva with isotonic saline-drenched gauze, the device was applied to the sublingual mucosa while avoiding pressure artefacts, and image sequences from at least 3 areas were recorded. Measurements were calculated before and immediately after the first and repeated exercise bout. Trained certified investigators used validated AVA v.3 software to blindly analyze the video clips, which were arranged in random order to avoid coupling. Expert recommendations [27] were followed for the quality and analysis of the recorded images.

The images were divided into four equal quadrants, and the flow in each was quantified by eye (0, no flow; 1, intermittent flow; 2, sluggish flow; 3, continuous flow) for each vessel diameter cohort (small, 10–20 µm; medium, 21–50 µm; large, 51–100 µm). The microvascular flow index (MFI) was calculated as the sum of each quadrant score divided by the number of quadrants in which the vessel type was visible. The final MFI was an average over a minimum of 12 quadrants (three regions, four quadrants per region) derived from the overall flow impressions of all vessels within a particular range of diameters in a given quadrant [35,36]. The total vessel density of the small vessels was calculated using the AVA software package with a cut-off diameter for small vessels (mostly capillaries) of <20 μm. Perfused small vessel density (PVD) was calculated as the number of crossings of perfused small vessels per the total length of three equidistant horizontal and vertical lines. The proportion of perfused small vessels (PPV) was defined by the percentage of crossings with perfused small vessels per the total length of three equidistant horizontal and vertical lines [22,27,36].

### 2.6. Evaluating the Glycocalyx: Measuring the Perfused Boundary Region

Visualizing human glycocalyx in vivo is extremely difficult due to its fragility, but where it is partially accessible to flowing red blood cells (RBC) at its luminal side is called the perfused boundary region (PBR) [37]. Destruction of the glycocalyx results in deeper penetration of the RBCs towards the endothelium and an increase in the PBR [18].

An SDF video microscope connected to a glycocalyx measurement system (GlycoCheck ICU^®^; Maastricht University Medical Center, Maastricht, The Netherlands) was used to visualize the sublingual microcirculation. Ten image sequences of 40 frames each were recorded in different positions and automatically analyzed to calculate the PBR. The RBC column was automatically measured in 3000 vascular segments. For each segment, 840 radial intensity profiles were obtained to measure the RBC column width, and the PBR was calculated as the distance from the median (P50) RBC column width to the (estimated) outer edge of the RBC-perfused lumen. As described elsewhere [38,39], the vessel segments were classified in 1 μm-wide diameter classes, and the median PBR values were determined for each diameter class before calculating the average PBR over a diameter range of 5 to 25 μm.

### 2.7. Evaluating the Glycocalyx: Serum Measurements of a Glycocalyx Damage Marker

A median antecubital vein was chosen for venipuncture. Blood serum was collected before and after SIE into vacuum tubes with a gel separator (5 mL; BD Vacutainer, Franklin Lakes, NJ, USA). The samples were allowed to clot, and the serum was separated by centrifugation (1200× *g*, 15 min) at room temperature. The serum samples were aliquoted and stored at −80 °C until analysis.

The concentration of syndecan-1 (Human Syndecan-1 (CD138) ELISA kit; BioVendor, Brno, The Czech Republic), a marker o glycocalyx shedding in the serum was determined with the corresponding ELISA kit following the instructions of the manufacturer.

### 2.8. Lactate and Glucose Concentration

The blood lactate concentration was measured in capillary blood. Blood samples (0.3 mL) taken from the fingertip before exercise and at 2 and 30 min after completing the exercise were analyzed for blood lactate concentrations using reagent strips (Lactate Pro; Arkray, Inc., Kyoto, Japan). The glucose concentration was measured at the same time (Glucocard X-mini Plus; Arkray, Japan).

### 2.9. Statistics

Baseline data for a power calculation were not available due to the novelty of the study. The number of volunteers in our explorative study was based on the sample size according to similar vascular studies [3]. Primary aim was to describe the changes in microcirculation and glycocalyx properties observed during provocative acute high-intensity exercise after two weeks of pomegranate supplementation.

Statistical analyses were performed with SPSS version 15.1 for Windows (SPSS Inc., Chicago, IL, USA). With respect to the small sample size, the data were analyzed with non-parametric tests and are presented as median values (25th to 75th percentiles). Differences between groups were tested with a Mann–Whitney *U* test. Friedman’s test was conducted to assess changes of quantitative parameters over multiple time points in each group followed by a Wilcoxon test to evaluate intragroup changes between two time points. A *p* value of <0.05 was considered significant. The magnitude of the effect of the intervention between groups was estimated by Cohen’s effect size (ES) test. In our study, the effect size was greater than 0.8 for all statistically significant results.

## 3. Results

### 3.1. The Baseline Characteristics

The baseline characteristics of the participants are presented in Table 1. There was no significant difference in age, body weight, or height between the groups. There were also no significant differences in the physiological parameters between the groups, which included the systemic hemodynamics, maximal oxygen uptake, glucose and lactate concentration, PBRs, microvascular flow and density, and syndecan-1 concentration.

### 3.2. Glycocalyx Characteristics and Microcirculation

No significant differences in the syndecan-1 concentration or PBRs were found between the groups at any point (Table 2). Overall, we did not find any significant differences in the sublingual microvascular flow or density parameters between the groups over the course of our research (Table 3; Figure 2).

### 3.3. Blood Sample Analysis

The lactate concentration was significantly higher after the first and repeat bouts in both groups. After the RB, it was also significantly higher in the pomegranate group (13.2 (11.9–14.8)) in comparison to the control group (10.3 (9.3–12.7) mmol/L; *p* = 0.017, ES = 1.4), as shown in Figure 3.

### 3.4. Cycling Power and Systemic Hemodynamic Parameters

During the study period, there were no significant differences in average cycling power or maximal heart rate between the groups in either the first or repeated bouts (FB and RB, respectively). The power outputs in the first and 12th SIE repetitions for the pomegranate and control groups during the FB were 800.7 (709.8–959.4) vs. 776.6 (660.3–820.4) W (*p* = 0.294) and 840.9 (659.0–988.8) vs. 719.9 (649.0–881.6) W (*p* = 0.401), while in the RB they were 810.1 (736.1–962.9) vs. 821.2 (700.2–854.0) W (*p* = 0.916) and 794.2 (709.6–966.2) vs. 762.0 (652.1–865.6) W (*p* = 0.529). Similarly, the maximal heart rate during the first and 12th SIE repetitions in the pomegranate and control groups during the FB were 153 (147–165) vs. 153 (142–168) beats/min (*p* = 0.958) and 167 (156–173) vs. 155 (144–167) beats/min (*p* = 0.126), respectively, while in the RB, they were 159 (151–171) vs. 151 (140–160) beats/min (*p* = 0.141) and 169 (154–176) vs. 161 (144–172) beats/min (*p* = 0.318).

We found that the heart rate and cardiac index (CI) were significantly higher after both bouts for both groups, while the MAP was significantly higher after the FB, but not after the RB. There were no significant differences between the control and pomegranate groups in heart rate, MAP, or CI (Table 4).

## 4. Discussion

To the best of our knowledge, this study is the first to investigate the influence of pomegranate extract supplementation on sublingual microcirculation and the glycocalyx during acute high-intensity sprint interval exercise. No significant differences in the glycocalyx, microcirculation flow, or density parameters were found over the course of the study. Acute sprint interval exercise bouts (12 × 5-s each interspaced with 3 min of active recovery) have been shown to boost an acute-phase response, resulting in post-exercise cytokine levels [25] similar to those observed during sepsis or inflammation [40]. They are expected to increase the level of oxidative stress, leading to glycocalyx damage and alterations in microcirculation.

It was demonstrated that the acute ingestion of 1000 mg pomegranate extract 30 min prior to high-intensity sprint exercise could enhance brachial artery diameter and blood flow, increasing the ergogenic effects. These improvements are likely due to the nitrate and polyphenol content in the pomegranate extract, which should enhance endothelial function and, as a consequence, NO production [3,41]. Polyphenols have been shown to activate endothelial NO synthase via phosphorylation. Furthermore, the protective performance of pomegranate against ROS-mediated NO scavenging may further explain the enhanced vasodilation after the exercise [11].

There is data that prolonged pomegranate consumption has resulted in lower blood pressure and improved endothelial function in clinical populations, but there has been a lack of data on pomegranate supplementation and changes in microcirculation during acute exercise.

There might be several reasons why there were no changes in microcirculation and glycocalyx found in this study. First, as it was mentioned above, opposite to other types of training, SIE is a type of exercise that does not cause injury to muscle fibers but induces metabolic stress [25]. As a consequence, being a provocative factor for the induction of microcirculatory alterations, it might act too weakly. Second, there are reports that exercise-induced improvements in blood pressure and flow-mediated dilation were prevented by antioxidant supplementation [42].

The participants in our study did not receive the pomegranate extract on the day the measurements were calculated. It should be noted that the dosage and regimen of supplementation might impact the result, as the peak concentration of ellagic acid in human plasma after the consumption of ellagitannins from pomegranate juice is reported to be 1 h after ingestion, with an elimination time of 4 h [43,44]. However, there is published work showing that a week after stopping pomegranate juice consumption (following two weeks of supplementation), the beneficial effects on the antioxidant status in the organism appear to remain [45].

Our study was performed in healthy young men, which may explain the lack of significant changes in heart rate and blood pressure between the groups [3]. These results confirm the findings of other researchers that acute high-intensity exercise that includes intensity differences has no apparent harmful effects on the glycocalyx or sublingual microcirculation in well-hydrated men [17,36].

Lactate has been proposed to estimate metabolic response to high-intensity exercise as it stands as a crucial glycolytically generated metabolite most likely released because of increased or accelerated anaerobic glycolysis and stress response. It serves as an essential buffer, enabling the production of significant amounts of energy rapidly when the oxidative capacity of the mitochondria is exceeded due to the increased rate of glucose metabolism [25]. We found that the lactate concentration was significantly higher after the initial and repeat bouts for both groups. It was significantly higher in the pomegranate group compared to the control group after the repeated bout, despite no differences between the groups in their average cycling power, maximal heart rate, and sublingual (central) microcirculation. Several mechanisms might explain this elevation in blood lactate concentration in the pomegranate group. For instance, the results correspond to those of a study conducted by Fuster-Munoz et al. [46] in which elevated extracellular K+ levels were found at the end of the study in participants who were supplemented with pomegranate juice. Since extracellular K+ most likely only accumulates to inhibitory levels at exercise intensities where lactic acid acidosis also occurs, the findings suggest that the accumulation of lactic acid might be a protective mechanism against muscle fatigue.

Exercise-induced ROS formation in humans is promoting insulin sensitivity [42]. Lactate metabolism in response to a glucose or insulin challenge in healthy subjects shows that the rate of glucose disposal is positively correlated with plasma lactate concentration. Consequently, the plasma lactate concentration increases under insulin-stimulating conditions [47]. However, despite the differences in lactate concentrations between the groups, there were no significant differences in the glucose concentrations over the course of our study. Interestingly, increase in insulin sensitivity, following physical exercise is completely abrogated by the daily ingestion of other commonly used antioxidants such as vitamin C and E [42].

Animal studies in young rat spinotrapezius muscle showed that acute antioxidant supplementation significantly reduced skeletal muscle blood flow both at rest and during contractions [48], possibly resulting in increased lactate production. Animal models have revealed that high dosages of antioxidant supplements could shut down specific redox-sensitive cell-signaling pathways and accordingly reduce the synthesis of new muscle mitochondria and endogenous antioxidant production [43], leading to impaired lactate clearance and oxidation [49].

The main limitation of the present study was the small sample size of each group. However, all participants in each group were homogeneous. Another limitation is that placebo was not used. It was done due to difficulties in developing a neutral material of similar color, consistency and taste as pomegranate extract. Moreover, there always remains the probability, that additional herbal material imitating spicy, fruity, astringent taste may impact changes in microcirculation. Researchers who analyzed the data of microcirculation were blinded and video images were randomized during the process of analysis to avoid bias. Only male participants were included in our study, and thus our results cannot readily be extrapolated to females. We also did not investigate oxidative stress markers, although microcirculation and glycocalyx status may mirror the oxidative stress status.

Further research is required to optimize the use of pomegranate derived supplements. A possible future research direction may be related to a different provocative factor in order to investigate whether it induces microcirculatory impairment via a higher level of oxidative stress. Another direction for future research might be aimed to investigate the mechanisms underlying the higher lactate level related to ingestion of antioxidants.

## 5. Conclusions

Two weeks of supplementation with pomegranate extract has no impact on changes in the microcirculation and glycocalyx during acute high-intensity sprint interval cycling-exercise. Thus, the reason for the pomegranate consumption-induced increase in blood lactate concentration remains unresolved. These results merit further investigation.

## Figures and Tables

**Figure 1 ijerph-17-04405-f001:**
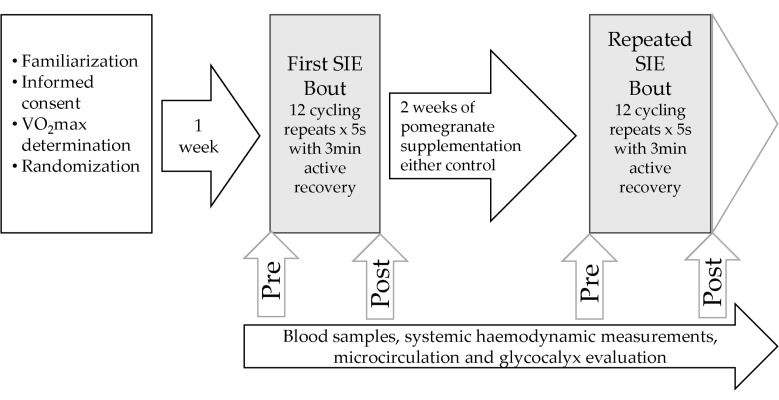
Experimental design.

**Figure 2 ijerph-17-04405-f002:**
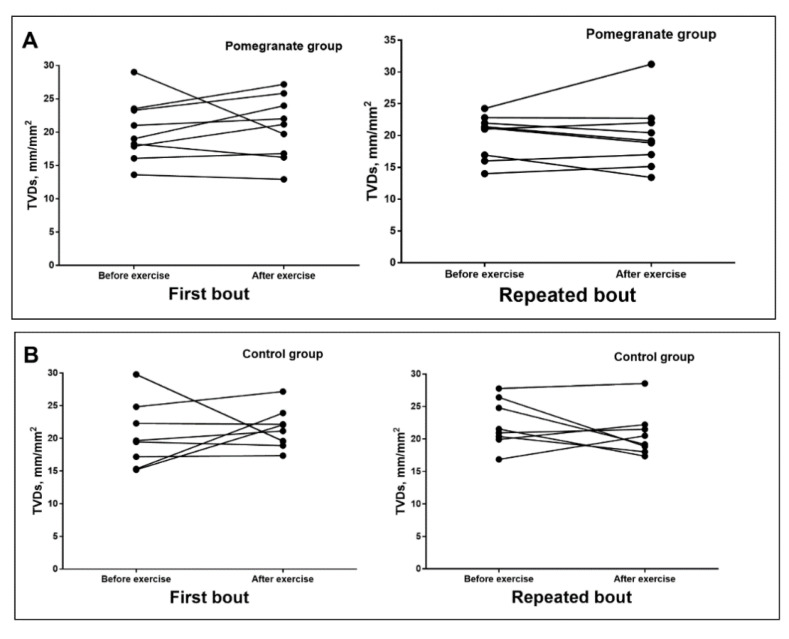
Change of total vessels density of small vessels (TVDs) in pomegranate (**A**) and control (**B**) groups.

**Figure 3 ijerph-17-04405-f003:**
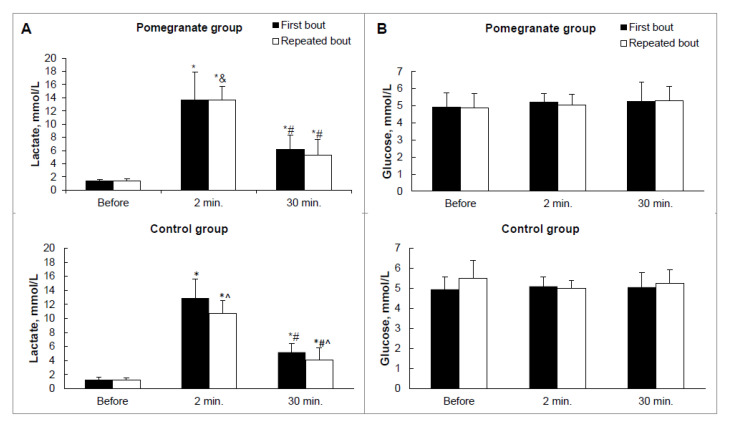
Lactate and glucose concentrations in pomegranate and control groups. (**A**) Lactate; (**B**) Glucose. * *p* < 0.05 compared with baseline values; # *p* < 0.05 between 2 min and 30 min; & *p* < 0.05 compared with control values; ^ *p* < 0.05 between first bout and repeated bout.

**Table 1 ijerph-17-04405-t001:** Baseline characteristics of the study participants.

Variables	Pomegranate Group	Control Group	*p*
Age (yr)	24 (22–25)	23 (22–28)	0.893
Mass (kg)	78.0 (68.2–84.6)	70.2 (65.0–81.1)	0.426
Height (cm)	180.0 (178.0–184.5)	180.0 (177.0–183.5)	0.929
Body mass index (kg/m^2^)	21.3 (20.7–27.3)	21.9 (19.8–25.3)	0.566
Fat (%)	11.9 (10.1–18.8)	11.2 (9.3–17.5)	0.700
VO_2_max (mL∙kg^−1^∙min^−1^)	45.9 (41.0–46.5)	42.2 (39.4–44.9)	0.172
Mean arterial pressure (mmHg)	94 (93–101)	97 (94–100)	0.756
Heart rate (beats/min)	76 (70–81)	79 (66–86)	0.690
Cardiac index (L/min/m^2^)	3.6 (3.1–4.4)	3.8 (2.9–4.4)	0.690
Lactate (mmol/L)	1.4 (1.3–1.6)	1.1 (1.0–1.9)	0.478
Syndecan-1 (ng/mL)	8.7 (2.7–34.7)	7.7 (1.7–16.8)	0.685
PBR (µm)	1.87 (1.63–2.05)	1.94 (1.66–2.00)	0.635
TVDs (mm/mm^2^)	19.0 (17.0–23.4)	19.7 (16.3–27.3)	0.222
PVDs (1/mm)	9.7 (8.8–13.6)	11.1 (9.7–15.7)	0.080
PPVs (%)	95.7 (93.0–99.3)	98.6 (95.3–98.8)	0.386
MFIs	3.00 (3.00–3.00)	3.00 (3.00–3.00)	0.955

**Table 2 ijerph-17-04405-t002:** Changes in glycocalyx parameters during the study period.

		Baseline		After 2 Weeks		
		Before	After	Before	After	*p*
**Syndecan-1 (ng/mL)**	PG	8.7(2.7–34.7)	7.1(4.0–31.9)	8.7(4.9–36.7)	9.3(4.7–37.5)	0.672
	Con	7.7(1.7–16.8)	7.4(1.7–17.2)	6.0(2.5–19.3)	5.8(4.7–16.5)	0.798
**PBR (µm)**	PG	1.87(1.63–2.05)	1.90(1.72–1.95)	1.97(1.81–2.10)	1.73(1.52–1.89)	0.563
	Con	1.94(1.66–2.00)	1.87(1.73–1.95)	1.82(1.63–1.98)	1.81(1.68–1.95)	0.496

Data are presented as the median (25th–75th percentiles). PBR: perfused boundary region. **p* < 0.05 compared with controls; ^a^
*p* < 0.05 compared with baseline.

**Table 3 ijerph-17-04405-t003:** Changes in microcirculatory parameters during the study period.

		Baseline		After 2 Weeks		
		Before	After	Before	After	*p*
**TVDs (mm/mm^2^)**	PG	19.0(17.0–23.4)	21.2(16.8–25.8)	21.4(16.9–22.8)	19.2(15.1–22.7)	0.932
	Con	19.7(16.3–27.3)	21.2(19.2–23.0)	21.5(20.2–27.1)	20.5(18.4–25.4)	0.978
**PVDs (1/mm)**	PG	9.7(8.8–13.6)	11.0(9.4–14.5)	11.9(9.3–12.4)	10.8(8.6–12.2)	0.508
	Con	11.1(9.7–15.7)	11.4(10.7–13.5)	12.0(10.4–15.9)	12.1(11.1–15.0)	0.996
**PPVs (%)**	PG	95.7(93.0–99.3)	96.1(93.8–98.8)	95.6(93.8–97.3)	95.6(95.0–100.0)	0.745
	Con	98.6(95.3–98.8)	96.4(91.4–98.9)	95.9(93.15–98.4)	95.8(91.7–98.3)	0.472
**MFIs**	PG	3.00(3.00–3.00)	3.00(2.84–3.00)	3.00(3.00–3.00)	3.00(3.00–3.00)	0.437
	Con	3.00(3.00–3.00)	3.00(2.81–3.00)	3.00(2.75–3.00)	3.00(2.81–3.00)	0.395

Data are presented as the median (25th–75th percentiles). TVDs: total vessel density of small vessels; PVDs: perfused vessel density of small vessels; PPVs: percentage of perfused small vessels; MFIs: microvascular flow index of small vessels. * *p* < 0.05 compared with controls; ^a^
*p* < 0.05 compared with baseline.

**Table 4 ijerph-17-04405-t004:** Changes in systemic hemodynamic parameters during the study period.

		Baseline		After 2 Weeks		
		Before	After	Before	After	*p*
**MAP (mmHg)**	PG	94(93–101)	107(96–116) ^a^	98(92–101)	104(94–107)	0.026
	Con	97(94–100)	103(99–115) ^a^	97(90–100)	100(95–107)	0.030
**HR (beats/min)**	PG	76(70–81)	106(96–116) ^a^	75(62–89)	104(91–116) ^a^	<0.001
	Con	79(66–86)	106(92–115) ^a^	77(63–84)	104(92–116) ^a^	<0.001
**CI (L/min/m^2^)**	PG	3.6(3.1–4.4)	4.4(3.5–4.9) ^a^	3.7(3.5–4.0)	4.5(3.7–4.7) ^a^	0.045
	Con	3.8(2.9–4.4)	4.2(3.4–5.1) ^a^	3.3(3.2–3.7)	3.9(3.5–4.9) ^a^	0.037

Data are presented as the median (25th–75th percentiles). MAP: mean arterial pressure; HR: heart rate; CI: cardiac index. * *p* < 0.05 compared with controls; ^a^
*p* < 0.05 compared with baseline.

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
