# Peer review of "Modulation of Endothelial Glycocalyx and Microcirculation in Healthy Young Men during High-Intensity Sprint Interval Cycling-Exercise by Supplementation with Pomegranate Extract. A Randomized Controlled Trial"

_ijerph, 2020, doi:10.3390/ijerph17124405_

Round 1

Reviewer 1 Report

Please see attached review=-=thanks.

Author Response

Dear Reviewer,

Thank you for the opportunity to improve the quality of the paper and the reviewers for their valuable input. This document contains a point-by-point answer to all the questions. In addition to that we have submitted a new version of the manuscript. Changes are depicted in red. We hope the answers will be satisfactory.

Sincerely yours,

Authors

Reviewer #1

General comments: The authors tested if pom extract ingested before a bout of sprint interval training had any effects upon the microcirculation, which for the most part, it did not. The sample size is small, and I question the authors’ use of sprint interval exercise, especially when no text is included regarding this. There are also parts of the method which are incomplete, which reduces my interest in this work.

R/ Thank you for the remark. We expanded the methods section. We agree that the sample size is small. The number of volunteers in our explorative study was based on the sample size according to similar vascular studies (1). The primary aim was to describe the changes in microcirculation and glycocalyx properties observed during provocative acute high-intensity exercise after pomegranate administration.

  1. Roelofs, E.J.; Smith-Ryan, A.E.; Trexler, E.T.; Hirsch, K.R.; Mock, M.G. Effects of pomegranate extract on blood flow and vessel diameter after high-intensity exercise in young, healthy adults. Eur J Sport Sci 2017, 17, 317-325.

Specific comments

Abstract, please list the dose of the pomegranate extract given; thank you.

R/ The dose was listed.

Please refer to the bout as sprint interval training throughout—high intensity exercise is quite vague, and I think in this section text is merited, albeit brief, denoting the traits of this exercise bout.

R/ We agree. The section was reconstructed.

Line 27 needs some data and/or p values for the changes in microcirculation and the glycocalyx related outcome—thank you.

R/ We agree. The data and p values were noted.

Introduction line 47: in whom is this ergogenic effect seen? Please list the specific population here and type of exercise completed in these studies.

R/ Thank you. The population and a type of exercise was listed.

Paragraph in lines 50-53 is too brief and seems out of place—it is recommended to revise this text or move it to some other place in this section.

R/ We agree. The section was reconstructed.

The end of the Introduction does now really “flow” from the preceding 2 paragraphs, specifically..a) this transition to the sublingual area is surprising; this needs to be introduced better; b) why the use of intense exercise here, and not something like MICT or strength training? This text does not lay any foundation for why interval exercise was performed rather than more traditional modes.

R/ Thank you for your insights. The section was reconstructed.

Method, is there a reason why the information regarding the extract is denoted first here, and not the participants or design of the study? And as I read this section, a Design section is needed.

R/ We agree. The method section was corrected.

What was the n in each group? How were men assigned to the control and experimental group? And why was a placebo group not used that ingested something that is inert?

R/ n = 9 in each group. Men were randomly assigned to the control and experimental group using an online randomization tool. Regarding the placebo, it is complicated to produce a placebo of taste, color, and odor of pomegranate extract. Moreover, additional herbal material imitating astringent taste may impact changes in microcirculation. Researchers who analyzed the data of microcirculation were blinded, and video images were randomized during the process of analysis to avoid bias.

Were these men regularly ingesting foods containing nitrates such as leafy vegetables? Was this even considered chronically or in the days leading up to each assessment?

R/ Thank you for your remark. The men were instructed not to eat foods containing nitrates during the study, and it was controlled daily.

VO2max measurement—how was attainment of VO2max identified? And why do you cite that people with a decent VO2max are the only ones able to do intense exercise? There are many studies in diabetics, older adults, obese individuals, all having VO2max much lower than your values, who have completed bouts of HIIE and SIE in prior literature. Ultimately, I see no need for this criterion as interval exercise is safe (Rognmo et al. 2008).

R/ VO2max was measured using the ramp exercise test. This test consists of a 4-minute warmup followed by a continuous increase in the speed of 0.1 km/h every 6 s until fatigue. A respiratory exchange ratio greater than 1.1, a maximum heart rate (HR) equal to 220 – age ± 10 beats per min, and a plateau in oxygen uptake with increasing workload (all the criteria had to be met) were used to verify that VO2max was achieved. Howley ET, Bassett DR Jr, Welch HG. Criteria for maximal oxygen uptake: review and commentary. Med Sci Sports Exerc. 1995;27(9):1292‐1301 Pulmonary gas exchange was analyzed using a portable analyzer (Oxycon Mobile; Jaeger, Hoechberg, Germany). Before each test, the equipment was calibrated according to the manufacturer’s recommendations. 

Section 2.5: do you have day-to-day reliability data? for all of these outcomes, especially the ones related to the microcirculation and the glycocalyx? Honestly, these are not outcomes widely assessed in most studies, so the reader needs to have a greater understanding of the reliability of these tests; thank you.

R/ Thank you for the accurate insight. So far, sublingual videomicroscopy has come in front as a gold standard for microcirculatory evaluation, and it is the most accurate way to evaluate tissue perfusion. This fact was added to the section. Researchers who analyzed data of microcirculation are certified to do it as all of them have completed an international course on microcirculation analysis, held by prof. C, Ince in the Microcirculation academy (Amsterdam, The Netherlands). The kappa coefficient for interrater variability was 0.86 (P < 0.001), and for intrarater variability was 0.79 (P < 0.001). The percentage of disagreement for the microvascular flow index in the sublingual part was 5.2%.

Cytocam IDF-imaging validation study: Aykut, Guclu et al. “Cytocam-IDF (incident dark field illumination) imaging for bedside monitoring of the microcirculation.” Intensive care medicine experimental vol. 3,1 (2015): 40. doi:10.1186/s40635-015-0040-7

What is syndekan 1? Please explain. And why were lactate and glucose measured? Are they related to these endothelial markers or to microcirculatory function? If not, they seem out of place and just variables that were “thrown in” to the study without rationale.

R/ Syndecan 1 is a proteoglycan used as a marker of glycocalyx degradation. Oxidative stress induces the shedding of syndecan-1 from the endothelial cell surface. It was added to the text. Lactate has been proposed to evaluate metabolic response as it is a critical glycolytically produced metabolite most likely released because of increased or accelerated anaerobic glycolysis and stress response. It acts as an essential buffer, allowing the production of significant amounts of energy rapidly when the oxidative capacity of the mitochondria is exceeded due to the increased rate of glucose metabolism.

What was the load used on the ergometer during SIE?

R/ Each bout of exercise comprised 12 repeats of 5 s (Verbickas et al., 2017) on a cycle ergometer (Monark, Ergomedic 894 EA, Vansbro, Sweden) interfaced with a laptop computer. Foot straps were used to secure the feet to the pedals. The pedal right arm crank starting position was 45◦ forward to the vertical axis. Upon the start command, the subject began pedaling and continued until the stop command. During cycling, when the subject had reached 100 rpm pedaling frequency, the computer automatically added the weight (7.5% of body mass). During the entire SIE, the subject received strong verbal encouragement.

Is there a reason why the extract was given as an absolute and not relative dose? This means that men with a higher mass get a lower dose.

R/ According to the studies daily, one or two glass of pomegranate juice is used for various studies on exercise performance. A spoonful (10 ml) of this concentrated pomegranate liquid thick extract had the antioxidant activity as over 200 ml of pomegranate juice. We chose the maximum dose (20 ml) to ensure the effect for all volunteers. Also, for practical purposes, it is more convenient to dose the same amount (especially for thick liquid extract).

Ammar, A., Bailey, S., Chtourou, H., Trabelsi, K., Turki, M., Hökelmann, A., & Souissi, N. (2018). Effects of pomegranate supplementation on exercise performance and post-exercise recovery in healthy adults: A systematic review. British Journal of Nutrition, 120(11), 1201-1216. doi:10.1017/S0007114518002696

I assume RM ANOVA was used to assess changes in your outcomes? Please denote this in your method section.

R/ Thank you. This was corrected. Differences between groups were tested with a Mann–Whitney U test. Friedman test was done to assess changes of quantitative parameters over multiple time points in each group followed by a Wilcoxon test to evaluate intragroup changes between two time points. A p value of <0.05 was considered significant.

Results can you please add subheadings to this section to better organize it? Thank you.

Your Method section does not denote that HR and PO were recorded during the SIE session—it needs to.

R/ We agree. The section was reconstructed and it was noted that HR and PO were recorded during the SIE session

It is awkward that following the PO data, you present data re. changes in BP and CI—in my view, these are tertiary outcomes and should be denoted after your glycocalyx and related microcirculation outcomes. These are listed in the title and comprise your main variables, so should be presented first here.

R/ We agree. The section was reconstructed.

Figure 2 seems to represent individual data—why is this merited? Individual responses to training are commonly reported, but without any rationale, I see no reason to do this for your outcomes.

R/ Thank you for insight. The sample size is not large so it was interesting to look at tendencies of individual response.

Discussion line 249 what kind of exercise bouts? It needs to be denoted here and in my view, only studies in which supramaximal bouts are used relate to your study and its findings. It all comes down to why SIE was used in your current study.

R/ We clarify the sentence in discussion” with proper citation. Acute sprint interval exercise bouts (12 × 5-s each interspaced with 3 min of active recovery) have been shown to boost an acute-phase response, resulting in post-exercise cytokine levels[...] similar to those observed during sepsis or inflammation [..]. They are expected to increase the level of oxidative stress, leading to glycocalyx damage and alterations in microcirculation.

Line 256: what kind of exercise performance? SIE is more reliant on anaerobic metabolism than most exercise modes—does this factor matter as to why no benefit of pom extract was shown in your study?

R/There might be several reasons why there were no changes in microcirculation and glycocalyx found in this study. First, as it was mentioned above, opposite to other types of training, SIE is a type of exercise that does not cause injury to muscle fibers but induces metabolic stress. As a consequence, being provocative factor for the induction of microcirculatory alterations, it might act too weak. Second, there are reports that exercise-induced improvements in blood pressure and flow-mediated dilation were prevented by antioxidant supplementation.

Lines 257-260: why is vitamin E/C pertinent to pom extract? What is the association b/w these data and your study?

R/ The main reason it was introduced is an antioxidative activity.

Line 269: ok, so there seem to be long lasting effects of pom extract on antioxidant status, but what about changes in your outcomes?

R/ Thank you for your insight. SIE did not induce changes in microcirculation and glycocalyx parameters, but we have no data about oxidative stress levels in our participants since antioxidant status, and markers of oxidative stress were not investigated in our study.

R/ A particularly substantial provocative factor and high level of oxidative stress are required to induce impairment of microcirculation. There might be several reasons why there were no changes in our outcomes: insufficient provocative factor. Supplementation may have failed to improve microcirculation because of possibly insufficient dose and a too short period of supplementation. However, previous studies on pomegranate effect on the level of oxidative stress have shown that a similar dose and period of supplementation should have an effect on antioxidant status after high-intensity training. There were no studies investigating the pomegranate effect on microcirculation. A more intense provocative factor in inducing impairment of microcirculation might be a perspective for future studies.

It was added to the limitations section.

Line 270: how much was dosed and who were the subjects and what exercise did they complete?

R/ We agree. Additional information was added in methods section.

Reviewer 2 Report

This is the manuscript entitled “Modulation of endothelial glycocalyx and microcirculation in healthy young men during high-intensity exercise by supplementation with pomegranate extract”. The present study explores the effect of chronic supplementation with pomegranate extract on  glycocalyx properties and microcirculation response to acute high-intensity exercise. The topic is interesting and novel, However, some more clarifications are needed:

Abstract:

You have to clarify from the abstract the daily dose of pomegranate. Please add height and body mass of the participants if possible BMI. Also specify the type/nature of the exercise bouts (cycling, running, strength  or other ?).

There no statistical results supporting your conclusion. Please add p values and effect size

Please modify ‘on changes to the microcirculation and glycocalyx” by “on microcirculation and glycocalyx responses to the acute high-intensity exercsie”

Introduction and discussion:

  • Ammar et al is a research group that have recently conducted a big number of studies assessing the ergogenic effects and beneficial physiological effects of Pomegranate supplementation during physical exercises. I was surprised that no one of their works was cited or discussed. Please consider citing the following study in L38 when you spoke about the ergogenic effect of POM (Pomegranate supplementation accelerates recovery of muscle damage and soreness and inflammatory markers after a weightlifting training session. PLOS one; 11(10):e0160305). Please also consider citing the following response in L40 when you spoke about the modulation of oxidative stress and inflammation (Effects of Pomegranate Juice Supplementation on Oxidative Stress Biomarkers Following Weightlifting Exercise. Nutrients; 9(8). pii: E819). In the discussion section: Please discuss the underlying mechanism of the effect of POM in the discussion section (the following systematic review can be helpful: Effects of pomegranate supplementation on exercise performance and post-exercise recovery: A systematic review. British Journal of Nutrition. 20(11):1201-1216).

  • Please also consider adding these references to your manuscript:
    Nutrients 2019, 11, 721; doi:10.3390/nu11040721
    Eur J Sport Sci. 2017 Apr; 17(3): 317–325.
    Nitric Oxide. 2006;15(2):93-102. doi: 10.1016/j.niox.2006.03.001.
    Nutrition. 2015; 32(5) DOI: 10.1016/j.nut.2015.11.002

  • In addition to the used polyphenol dose, it is widely accepted that the bioavailability of the ingested phenolic compounds is an important factor influencing the health effect of the ingested supplementation. Variation in phenolic bioavailability ranges from <1% for anthocyanins, (less absorbed) to (≈34%) for isoflavones (well absorved). This can be one additional point in the discussion section.

Methods:

  • Physically active: you mean recreationally active or they are trained population ? please specify.
  • An experimental design figure would help understand the protocol.
  • Please Specify also the used design parallel group ?, blinding ? etc.
  • In the abstract it was stated “the two exercise bouts”, there is only a protocol for one exercise bouts in the method section, please clarify.
  • The exact composition of the supplement should be provided (caloric sugar ,mineral etc)
  • How POM supplements were prepared and how they was given (capsule or …). Because of the different parts that are eaten, pomegranate’s effects are also different. The author should specify the medicinal parts used to prepare the Pomegrante supplementation. For example: did supplementation come from the peel, aril, seeds of pomegranate, or from the whole fruit (especially that each part have a different phenolic content and antioxidant capacity).
  • The allowed physical activity during the supplementation period should be specified (as performing different amount of exercises in 2weeks period may create an adaptation process and/or repeated boot effect and affect the performance and the physiological responses in the post test). Did both groups perform similar activities and did the dietary intake, nature of food (antioxidant etc,,) was controlled during the supplementation period for both group, what about a standardized meal the day of the dosage ?
  • The exact statistical test (2way Anova or other ?) should be specified with the levels 2 (..) x 2 (..) ??

Results

  • Effect size should also be provided.

Minor:

  • L69-70: please add a reference
  • L112 “To minimize the effect of diurnal biological variations, each subject was tested at the same time of day”. Please add the following ref (Effect of time-of-day on biochemical markers in response to physical exercise. J Strength Cond Res. 31(1):272-282)

Author Response

Dear Reviewer,

Thank you for the opportunity to improve the quality of the paper and the reviewers for their valuable input. This document contains a point-by-point answer to all the questions. In addition to that we have submitted a new version of the manuscript. Changes are depicted in red. We hope the answers will be satisfactory.

Sincerely yours,

Authors

Reviewer #2

Abstract:

You have to clarify from the abstract the daily dose of pomegranate. Please add height and body mass of the participants if possible BMI. Also specify the type/nature of the exercise bouts (cycling, running, strength  or other ?).

R/ The daily dose of pomegranate was clarified. The abstract section was reconstructed.

There no statistical results supporting your conclusion. Please add p values and effect size

R/ Thank you for your remarks. p values were added.

Please modify ‘on changes to the microcirculation and glycocalyx” by “on microcirculation and glycocalyx responses to the acute high-intensity exercsie”

R/ It was changed. Thank you.

Introduction and discussion:

Ammar et al is a research group that have recently conducted a big number of studies assessing the ergogenic effects and beneficial physiological effects of Pomegranate supplementation during physical exercises. I was surprised that no one of their works was cited or discussed. Please consider citing the following study in L38 when you spoke about the ergogenic effect of POM (Pomegranate supplementation accelerates recovery of muscle damage and soreness and inflammatory markers after a weightlifting training session. PLOS one; 11(10):e0160305). Please also consider citing the following response in L40 when you spoke about the modulation of oxidative stress and inflammation (Effects of Pomegranate Juice Supplementation on Oxidative Stress Biomarkers Following Weightlifting Exercise. Nutrients; 9(8). pii: E819). In the discussion section: Please discuss the underlying mechanism of the effect of POM in the discussion section (the following systematic review can be helpful: Effects of pomegranate supplementation on exercise performance and post-exercise recovery: A systematic review. British Journal of Nutrition. 20(11):1201-1216).

Please also consider adding these references to your manuscript:
Nutrients 2019, 11, 721; doi:10.3390/nu11040721
Eur J Sport Sci. 2017 Apr; 17(3): 317–325.
Nitric Oxide. 2006;15(2):93-102. doi: 10.1016/j.niox.2006.03.001.
Nutrition. 2015; 32(5) DOI: 10.1016/j.nut.2015.11.002

R/ Thank you. The introduction and discussion sections were reconstructed.

In addition to the used polyphenol dose, it is widely accepted that the bioavailability of the ingested phenolic compounds is an important factor influencing the health effect of the ingested supplementation. Variation in phenolic bioavailability ranges from <1% for anthocyanins, (less absorbed) to (≈34%) for isoflavones (well absorved). This can be one additional point in the discussion section.

R/ Thank you for the accurate note.

Bioactive plant compounds such as polyphenols, are relatively poorly absorbed, the absorption ranging from 0.3% to 43%, and the circulating plasma concentrations of their metabolites can be low. Phenolic acids, flavonoids, and tannins are present in different parts of pomegranate fruit and the studies demonstrated that combinations of pomegranate extracts from different parts of the fruit were more effective than a single extract (1). Our liquid thick extract includes the full spectrum of all polyphenols in the fruit red seeds and also in the white membranes containing these seeds. In discussion section we were concentrated on the impact to microcirculation.  

1.Manach C, Williamson G, Morand C, Scalbert A, Remesy C. Bioavailability and bioefficacy of polyphenols in humans. I. Review of 97 bioavailability studies. Am J Clin Nutr 2005; 81: (1 Suppl.):230S–242.

Methods:

Physically active: you mean recreationally active or they are trained population ? please specify.

R/ Thank you for the remark. Men were recreationally active. It was added.

An experimental design figure would help understand the protocol.

R/ We agree. The figure was added.

Please Specify also the used design parallel group ?, blinding ? etc.

R/ Thank you. The method section was reconstructed.

In the abstract it was stated “the two exercise bouts”, there is only a protocol for one exercise bouts in the method section, please clarify.

R/ Thank you for the remark. It was clarified.

The exact composition of the supplement should be provided (caloric sugar ,mineral etc)

R/ We add exact composition to section 2.3 Experimental design and supplementation

Average nutritional values for 100 ml of pomegranate supplement: fat < 0,5 g, carbohydrate 47 g, protein 0,8 g, salt 0,04 g, potassium 1500 mg.

How POM supplements were prepared and how they was given (capsule or …). Because of the different parts that are eaten, pomegranate’s effects are also different. The author should specify the medicinal parts used to prepare the Pomegrante supplementation. For example: did supplementation come from the peel, aril, seeds of pomegranate, or from the whole fruit (especially that each part have a different phenolic content and antioxidant capacity).

R/ Thank you for the accurate note. We inserted additional points to 2.3 section.

Pomegranate supplement was obtained through a low heat reduction of pomegranate pulp, concentrated under vacuum, freeze-dried to preserve its properties, and then fermented. The extract contains the full spectrum of polyphenols out of the red fruit seeds and the white membranes of the seeds, which are typically not consumed. Liquid thick pomegranate extract was provided in a graded cup.

The allowed physical activity during the supplementation period should be specified (as performing different amount of exercises in 2weeks period may create an adaptation process and/or repeated boot effect and affect the performance and the physiological responses in the post test). Did both groups perform similar activities and did the dietary intake, nature of food (antioxidant etc,,) was controlled during the supplementation period for both group, what about a standardized meal the day of the dosage ?

R/ Thank you for the remarks. Both groups were instructed not to consume food containing nitrates. Physical activity of men was similar in both groups. They did not perform any exercises during the period of the study. It was controlled on daily basis.

The exact statistical test (2way Anova or other ?) should be specified with the levels 2 (..) x 2 (..) ??

R/ Thank You. Statistical analysis part was updated.

Results

Effect size should also be provided.

R/ Effect size of the interventions could not be estimated due to the novelty of the study. The number of volunteers in our explorative study was based on the sample size according to similar vascular studies (1).

  1. Roelofs, E.J.; Smith-Ryan, A.E.; Trexler, E.T.; Hirsch, K.R.; Mock, M.G. Effects of pomegranate extract on blood flow and vessel diameter after high-intensity exercise in young, healthy adults. Eur J Sport Sci 2017, 17, 317-325.

Minor:

L69-70: please add a reference

R/ The reference was added

L112 “To minimize the effect of diurnal biological variations, each subject was tested at the same time of day”. Please add the following ref (Effect of time-of-day on biochemical markers in response to physical exercise. J Strength Cond Res. 31(1):272-282)

R/ The reference was added.

Author Response

Dear Reviewer,

Thank you for the opportunity to improve the quality of the paper and the reviewers for their valuable input. This document contains a point-by-point answer to all the questions. In addition to that we have submitted a new version of the manuscript. Changes are depicted in red. We hope the answers will be satisfactory.

Sincerely yours,

Authors

Reviewer #3

Abstract

The methods are unclear; so, the authors should improve this part (e.g. dosage and timing of pomegranate supplement, more detail on the HIIT protocol).

R/ We agree. The methods section was corrected.

Line 31- I don’t think that supplementing for only two weeks considered a chronic supplementation. Please remove “chronic” here.

R/ Thank you for the remark. “Chronic” was removed.

Introduction

Line 39- ….”The natural components of pomegranate, including nitrate and 38 polyphenols, may be of additional benefit for endothelial function……” A reference is required here for this statement.

 R/ Thank you. The reference was added.

Line 50-52 - “In a comparison of pomegranate juice and extract, the extract reduced platelet activation to a larger extent for better endothelial function and was a more powerful antioxidant [7]. However, its 51 effect on microcirculation and the glycocalyx during acute high-intensity exercise has not been 52 studied.” This paragraph should be moved down to the last part of the introduction as it doesn’t fit where it is now.

R/ Thank You for the remark. The paragraph was moved.

Line 64-65- ….”The sugar protein glycocalyx, a gel-like layer, coats all healthy vascular endothelium on the luminal side, providing a micro-environment for many important vascular processes….” A reference is required here for this statement.

 R/ Thank you. The reference was added.

Line 65- Add “for example” before this line “Glycocalyx regulates vascular permeability, influences blood cell-vessel wall interactions, affects the rheology, and controls the microenvironment”.

R/ Thank you, the line was corrected.

Line 69 -It is widely known that regular training is beneficial for health and might help to prevent or delay cardiovascular, metabolic, and other chronic diseases. A reference is required here for this statement.

 R/ Thank you. The reference was added.

Line 111 “2.4. Pre-supplementation” I would suggest changing the heading to something for example, Familiarization session.

R/ Thank you. The heading was changed.

Line 131- Change “The subject” to “The subjects”

R/ Thank you, it was corrected.

Line 136 “subject was instructed to perform with maximum effort from the beginning of the test until instructed to stop [24].” Were there any verbal encouragements provided here? If yes. Please state that.

 R/ Thank you. There were verbal encouragements provided. We have stated it in an article.

Line 145 “2.6. Evaluating Microcirculation” Despite a clear and through methodology here, the exact timing of this evaluation is not clear. Please ensure that you state that.

 R/ We agree. The timing was added.

Line 190- More details on ELISA kit should be provided here. For example, the name of company, Inter and/or Intra- assay intervals. What was the timing of this measurement? Was it immediately after the test? How the serum was serrated? All the detail should be included here.

R/ Thank you for the remark. Additional information was included.

Line 198- The primary outcome was the sublingual total vessel density of small vessels. Please remove this.

 R/ Thank you for remark. It was removed

Throughout the manuscript, there is inconsistencies in levels or concentrations for glucose and lactate. Please consider changing them all to “concentrations”.

 R/ Thank you, it was done.

Discussion

Overall, the discussion fails to thoroughly the existing literature and how/why the outcomes of this study would support or perhaps differ from them.

Line 190 – Since the main goal of this study was to evaluate glycocalyx properties and microcirculation, these results must be comprehensively discussed in this part.

R/ We agree. The section was reconstructed.

Line 257-258- “…Nikolaidis et al. [34] published a meta-analysis on 11 research studies regarding the efficacy of vitamin C and E supplementation…” Why would you refer to a study investigating another type of antioxidants, given that we already know the mechanisms of actions may not be similar to pomegranate. I would suggest removing this and instead, discuss more papers which directly used any form of pomegranate in their interventions.

R/ We agree. It was removed.

What is the practical application of this project? What are the possible future research directions?

R/ Our results showed that prolonged supplementation with pomegranate extract has no impact on changes in the microcirculation and glycocalyx during acute high-intensity exercise. However, a substantial provocative factor and a high level of oxidative stress are required to induce it. A possible future research direction may be related to a different provocative factor in order to investigate whether it induces microcirculatory impairment. Another direction for future research might be aimed to investigate mechanism of higher lactate level related to ingestion of antioxidants.

Reviewer 4 Report

Title and Abstract:

  • Identification as a randomized trial in the title
  • The first person appears. A scientific article should not be written in the first person, it should always be written in an impersonal manner. (Line 19)

Keywords: The keywords must not be the same as those in the title and should be ordered alphabetically.

Introduction:

  • The first person appears. A scientific article should not be written in the first person, it should always be written in an impersonal manner. (Line 83)
  • The importance of conducting this study should be reflected
  • The general objective should appear at the end of the introduction. The objective should be clearly written, referring to the population, the intervention, the comparison and the results (PICO strategy)

Methods:

  • As a clinical trial, the study should follow the “Consort Statement”, which comprises a checklist with a minimum set of evidence-based recommendations for reporting randomized trials and a flow diagram.
  • The methodology should begin with the description of the trial design
  • The registration number and name of the trial register should be indicated
  • You must indicate the eligibility criteria for the participants, where they were from (also which was the local ethics committee: indicate name), and where the data were collected.
  • How sample size was determined? You must indicate the method and type used to generate the sequence of random assignment of participants. In addition, you should indicate who generated the random allocation sequence, who enrolled participants, and who assigned participants to interventions

Results:

  • For each group, the losses and exclusions after randomization should be indicated, along with the reasons, in a flow diagram.
  • The first person appears. A scientific article should not be written in the first person, it should always be written in an impersonal manner. (Line 218,231,232…)

Discussion:

  • The first person appears. A scientific article should not be written in the first person, it should always be written in an impersonal manner. (Line 246,248…)
  • Future research directions should be mentioned

General commentary for authors

The study is very well planned and written, just add and correct some aspects, especially in methodology, to obtain a better quality of it. The Consort Statement must be properly followed.

Author Response

Dear Reviewer,

Thank you for the opportunity to improve the quality of the paper and the reviewers for their valuable input. This document contains a point-by-point answer to all the questions. In addition to that we have submitted a new version of the manuscript. Changes are depicted in red. We hope the answers will be satisfactory.

Sincerely yours,

Authors

Title and Abstract:

Identification as a randomized trial in the title

The first person appears. A scientific article should not be written in the first person, it should always be written in an impersonal manner. (Line 19)

R/ We agree, it was corrected.

Keywords: The keywords must not be the same as those in the title and should be ordered alphabetically.

R/ Thank you, it was changed.

Introduction:

The first person appears. A scientific article should not be written in the first person, it should always be written in an impersonal manner. (Line 83)

R/ We agree, it was corrected.

The importance of conducting this study should be reflected

R/ We agree. There were changes made in introduction section.

The general objective should appear at the end of the introduction. The objective should be clearly written, referring to the population, the intervention, the comparison and the results (PICO strategy)

R/ Thank you. The objective was added.

Methods:

As a clinical trial, the study should follow the “Consort Statement”, which comprises a checklist with a minimum set of evidence-based recommendations for reporting randomized trials and a flow diagram. The methodology should begin with the description of the trial design. The registration number and name of the trial register should be indicated

R/ Thank you for remarks. The methods were reconstructed following the “Consort statement”.

You must indicate the eligibility criteria for the participants, where they were from (also which was the local ethics committee: indicate name), and where the data were collected.

R/ We agree. The methods section was reconstructed. Volunteers were mostly students of universities in Kaunas city, Lithuania.

How sample size was determined? You must indicate the method and type used to generate the sequence of random assignment of participants. In addition, you should indicate who generated the random allocation sequence, who enrolled participants, and who assigned participants to interventions

R/ We based the number of volunteers in our explorative study on the sample size according to earlier vascular studies. Primary aim was to describe the changes in microcirculation and glycocalyx properties observed during provocative acute high-intensity exercise after pomegranate administration. The participants were randomly assigned to the experimental and control groups using online randomization tool. (https://www.randomizer.org/)

Results:

For each group, the losses and exclusions after randomization should be indicated, along with the reasons, in a flow diagram.

R/ There were no losses and exclusions after randomization.

The first person appears. A scientific article should not be written in the first person, it should always be written in an impersonal manner. (Line 218,231,232…)

R/ Thank you. It was corrected.

Discussion:

The first person appears. A scientific article should not be written in the first person, it should always be written in an impersonal manner. (Line 246,248…)

R/ Thank you. It was corrected.

Future research directions should be mentioned

R/ We agree. It was mentioned.

Round 2

Reviewer 1 Report

I appreciate the great lengths that the Authors took to rewrite a lot of the manuscript text, and in addition, construct a very complete rebuttal.  The paper is now more clear and the rationale for the study has been strengthened, which were not the case in the initial iteration.

Author Response

I appreciate the great lengths that the Authors took to rewrite a lot of the manuscript text, and in addition, construct a very complete rebuttal.  The paper is now more clear and the rationale for the study has been strengthened, which were not the case in the initial iteration.

R/ Thank you. Your previous comments were very helpful and let us to improve the quality of the manuscript.

Reviewer 2 Report

i would like to than the authors for theie efforts in revising the manuscript

Minor comment: 

You added in the statistics : Effect size of the interventions could not be estimated due to the novelty of the study. The number of volunteers in our explorative study was based on the sample size according to similar vascular
studies [3]. Primary aim was to describe the changes in microcirculation and glycocalyx properties observed during provocative acute high-intensity exercise after pomegranate supplementation.

My suggestion was about adding effect size such as d cohen (e.g., if t-test, mann whitney were used) or eta squared (e.g., if Anova was used). 

However your response (could not be estimated due to the novelty of the study. The number of volunteers in our explorative study was based on the sample size according to similar vascular
studies [3].) seems to be more related to sample size not effect size.

I would prefer to delete the newly added paragraph in the statsics session and just the move this sentence (The number of volunteers in our explorative study was based on the sample size according to similar vascular
studies [3]) to the beggining of the participants section.

I hope you can provide non-parametric effect sizes with the p values calculated from Mann–Whitney U test, Friedman’s test or Wilcoxon

Author Response

Thank you for the opportunity to improve the quality of the paper and the reviewers for their valuable input. This document contains a point-by-point answer to all the questions. In addition to that we have submitted a new version of the manuscript. Changes are depicted in red.

Sincerely yours,

Authors

R/ We agree. We added effect size for lactates in the results. We added an explanation about Cohens d in the statistic section. Also, we changed the first sentences in the statistics section. Lines 236 and 245-247

In our study, the effect size was greater than 0.8 for all statistically significant results.

Reviewer 3 Report

Dear authors,

I would like to thank you for addressing my comments on the manuscript. I believe that the quality of manuscript has significantly improved which makes it fit for publication. However, I still have some minor comments that should be taken care of. 

  1. I understand that you have replaced "chronic with "prolong", but why don't use 2-week instead to make sure it conveys the message accurately.
  2. well, you have used "high-intensity sprint interval cycling-exercise in your study, so this should be presented in the title rather than "high intensity exercise.  
  3. line 90- "during acute high-intensity exercise in young healthy men compared to no supplementation." change " high-intensity exercise" to "sprint interval cycling-exercise".
  4. line 106- Table 1. Baseline characteristics of the study participants. Please change "controls" to "control group" in the table.
  5.  

Author Response

Thank you for the opportunity to improve the quality of the paper. This document contains a point-by-point answer to all the questions. In addition to that we have submitted a new version of the manuscript. Changes are depicted in red.

I understand that you have replaced "chronic with "prolong", but why don't use 2-week instead to make sure it conveys the message accurately.

R/ Thank you, it was changed: Line 35, 89, 388

well, you have used "high-intensity sprint interval cycling-exercise in your study, so this should be presented in the title rather than "high intensity exercise.

R/ Thank you, it was added to the title.

line 90- "during acute high-intensity exercise in young healthy men compared to no supplementation." change " high-intensity exercise" to "sprint interval cycling-exercise".

R/ Thank you, it was changed.

line 106- Table 1. Baseline characteristics of the study participants. Please change "controls" to "control group" in the table.

R/ Thank you, it was changed.

Reviewer 4 Report

Methods:

  • The Consort statement has not been followed through.
  • The section of the study design should appear at the beginning.
  • Following the Consort statement, the table showing the baseline characteristics of the participants should appear in results, not in methods.
  • The section on participants in general and the criteria for selecting participants, in particular, should be better explained.
  • The registration number and the name of the test register have not been indicated.

Author Response

Thank you for the opportunity to improve the quality of the paper. This document contains a point-by-point answer to all the questions. In addition to that we have submitted a new version of the manuscript. Changes are depicted in red.

Sincerely yours,

Authors

  • The Consort statement has not been followed through.
  • The section of the study design should appear at the beginning.

R/ Thank you, it was moved.

  • Following the Consort statement, the table showing the baseline characteristics of the participants should appear in results, not in methods.

R/ Thank you, it was moved.

  • The section on participants in general and the criteria for selecting participants, in particular, should be better explained.

R/ We agree, the section was reconstructed. Lines 133-135.

  • The registration number and the name of the test register have not been indicated.

R/ The registration numbers and the names of the local research ethics committees are provided in the lines 142-144